# Comparative Analysis of Volatile Constituents in Root Tuber and Rhizome of *Curcuma longa* L. Using Fingerprints and Chemometrics Approaches on Gas Chromatography–Mass Spectrometry

**DOI:** 10.3390/molecules27103196

**Published:** 2022-05-17

**Authors:** Guang-Mei Tang, Yi-Ting Shi, Wen Gao, Meng-Ning Li, Ping Li, Hua Yang

**Affiliations:** 1State Key Laboratory of Natural Medicines, China Pharmaceutical University, No. 24 Tongjia Lane, Nanjing 210009, China; tangguangmei2022@163.com (G.-M.T.); erin643747964@163.com (Y.-T.S.); gw_cpu@126.com (W.G.); 15251845246@163.com (M.-N.L.); 2State Key Laboratory Southwestern Chinese Medicine Resources, Chengdu University of Traditional Chinese Medicine, Chengdu 611137, China

**Keywords:** *Curcuma longa* L., volatile oil, bisabolane-type sesquiterpene, fingerprint, antioxidant

## Abstract

The root tuber and rhizome of *Curcuma longa* L., abbreviated, respectively, as RCL and RHCL, are used as different medicines in China. In this work, volatile oils were extracted from RCL and RHCL. Then, gas chromatography–mass spectrometry (GC–MS) was used for RCL and RHCL volatile oils analysis, and 45 compounds were identified. The dominant constituents both in volatile oils of RCL and RHCL were turmerone, (−)-zingiberene, and *β*-turmerone, which covered more than 60% of the total area. The chromatographic fingerprint similarities between RCL and RHCL were not less than 0.943, indicating that their main chemical compositions were similar. However, there were also some compounds that were varied in RCL and RHCL. Based on the peak area ratio of 45 compounds, the RCL and RHCL samples were separated into principal component analysis (PCA) and partial least squares discriminant analysis (PLS-DA). Then, 20 compounds with a variable importance for the projection (VIP) value of more than 1 were the high potential contributors for RCL and RHCL differences. Furthermore, ferric ion-reducing antioxidant power (FRAP) assay results demonstrated that the volatile oils of RCL and RHCL had antioxidant activities. This study provided the material basis for the research of volatile components in RCL and RHCL and contributed to their further pharmacological research and quality control.

## 1. Introduction

Traditional Chinese medicines (TCMs) are receiving worldwide attention due to their low side effects and good therapeutic efficacies. Many TCMs are from different parts of the same origin plants, such as Nelumbinis Semen and Nelumbinis Receptaculum [1] and Isatidis Radix and Isatidis Folium [2]. *Curcuma longa* L., an herbaceous plant from the *Zingiberaceae* family, is popular worldwide in the food, cosmetics, textile, and pharmaceutical industries. Multiple parts of *Curcuma longa* L. are used as medicines. As recorded in *Chinese Pharmacopoeia* (2020 edition), the root tuber of *Curcuma longa* L. (RCL) can promote blood circulation and relieve pain, promote qi and relieve depression, clear heart and cool blood, and treat hepatitis with jaundice [3]. Modern pharmacological research has shown that it has a variety of biological activities including improving the blood stasis model [4,5], alleviating pain [6], anti-tumor activities [7], et al. The rhizome of *Curcuma longa* L. (RHCL) is known as *Jianghuang* in Chinese. Its actions include promoting qi, breaking blood, unblocking the meridian, and relieving pain [3]. It has been used to treat various diseases, such as Alzheimer’s disease (AD) [8,9,10], diabetes [11,12,13], cancer [14,15,16], liver disease [17,18,19], cardiovascular disease [20,21,22,23,24], et al. Apparently, the efficacies of RCL and RHCL are somewhat different, which may be caused by their chemical differences. Chemical components in *Curcuma longa L*. contain curcuminoids, volatile oils, alkaloids, microelements, and other components. In the past, most research mainly focused on curcuminoids. However, many studies had shown that the volatile oils, another main component of *Curcuma longa* L., had important activities including anti-inflammatory [25], anti-tumor [26], anti-thrombosis [27], stroke improvement [28], anti-fungal [29], anti-aflatoxigenic [29,30], diabetes improvement [31], anti-hyperlipidaemic [32], et al. These indicated that the volatile oils also have the potential to characterize the quality of *Curcuma longa* L.

Gas chromatography–mass spectrometry (GC–MS) has widely been used to analyze volatile compounds, owing to its integrated superiorities of excellent separation power, highly sensitive detection, and improved identification based on sufficient ion information [33,34]. Chromatographic fingerprints were commonly applied to the holistic quality assessment of TCMs [35]. Generally, similarity analysis of fingerprints is used to assess the consistency of TCMs. Equally important, identifying the chemical differences contributes to discriminating the herbal quality variance across different samples. However, the subtle chemical differences are usually concealed under the holistic consistencies of their chromatograms [36]. Therefore, chemometric techniques and pharmacological activity were introduced to evaluate the quality. For example, principal component analysis (PCA) and partial least squares discriminant analysis (PLS-DA) were widely used to distinguish the different herbs and accurately group samples.

In this work, the GC–MS fingerprints of the total volatile oils extracted from RCL and RHCL were established. Chemical constituents in the volatile oils were characterized and compared using PCA and PLS-DA on their peak area ratio. Additionally, the antioxidant activities of total volatile oils from RCL and RHCL were evaluated and compared using the ferric ion-reducing antioxidant power (FRAP) method. Through this analysis, we expect to provide a more material basis to improve the comprehensive quality evaluation of RCL and RHCL.

## 2. Results and Discussions

### 2.1. Characterization of Compounds in RCL and RHCL Volatile Oils by GC–MS

The volatile oils of *Curcuma longa* have a large variety of pharmacological properties. Therefore, the volatile oils of RCL and RHCL were extracted by hydrodistillation according to *Chinese Pharmacopoeia* and then analyzed by GC–MS. The extracted yields of RHCL were more than 5%, which was much higher than that of RCL, which was on average 1.8%. The total ion chromatograms of RCL and RHCL volatile oils are shown in Figure 1. By comparing the obtained mass spectra data with the NIST MS spectra database and previous literature data [6,33,37,38,39], we totally identified 45 compounds, mainly including monoterpenes, sesquiterpenes, and other types. Their chemical structures are shown in Figure 2, and the compounds’ information and their peak area ratio in all tested samples are listed in Table 1. Among them, 14 compounds (peaks 4, 5, 6, 7, 8, 10, 12, 14, 15, 16, 18, 20, 34, and 39) were further confirmed with reference standards. For RCL volatile oils, the main components were turmerone (42.78%), (−)-zingiberene (11.99%), *β*-turmerone (10.69%), *β*-sesquiphellandrene (6.17%), *ar*-turmerone (5.58%), caryophyllene (3.42%), and terpinolene (2.67%), while the main constituent of RHCL included turmerone (35.18%), (−)-zingiberene (16.20%), *β*-turmerone (10.48%), *β*-sesquiphellandrene (9.06%), *ar*-turmerone (8.93%), *α*-curcumene (2.28%), and terpinolene (1.93%). The top five highest concertation constituents in RCL were the same as RHCL, all of them belonging to bisabolane-type sesquiterpene, which has many differences with other herbal medicines from *Curcuma* [40,41]. Accordingly, the first three, i.e., turmerone, (−)-zingiberene, and *β*-turmerone, covered more than 60% both in the RCL and RHCL volatile oil. Furthermore, we found that all of the samples in our research belonged to the reported major chemical type of turmeric in Xu’s article [42]. In all, the results showed that the main volatile components of RCL and RHCL were basically the same, only their relative contents were somewhat different.

### 2.2. GC–MS Fingerprint and Similarity Analysis

#### 2.2.1. Methodology Validation of Fingerprint Analysis

In order to develop the GC–MS fingerprint of the RCL and RHCL volatile oils, the precision, stability, and repeatability of the analytical method were assessed. The chromatographic similarity between the six repeated injections of the same sample and their common chromatography were not less than 0.999. The obtained chromatograms of one sample solution at 0, 2, 4, 8, 12, 20, and 24 h showed similarity with their common chromatography at 1.000. Moreover, the chromatographic similarity of six independent sample solutions from the same volatile oil was 1.000. Furthermore, the relative standard deviation (RSD) values of the relative retention time (RRT) and relative peak area (RPA) of 11 characteristic peaks (analyte/IS) were calculated (Table 2). The precision did not exceed 0.03% for RRT and 4.68% for RPA. The stabilities of RRT and RPA were not more than 0.04% and 3.65%, respectively. The results of the repeatability were not more than 0.04% for RRT and 5.07% for RPA. All the results showed that the instruments and methods were valid and suitable for analysis.

#### 2.2.2. Establishment of GC–MS Fingerprints and Similarity Analysis

The GC–MS chromatographic data of six batches RCL and six batches RHCL volatile oil were analyzed and imported into the Similarity Evaluation System for Chromatographic Fingerprint of Traditional Chinese Medicine software version 2004 A (Chinese Pharmacopoeia Commission, Beijing, China). Then, the fingerprints were obtained using the median method with a time width of 0.1 (Figure 3). The peaks found in all samples with good resolution and intensity were specified as common peaks. As shown in Figure 3, we identified seventeen common peaks. Their peak number was consistent in Table 1. These peaks included the main and characteristic peaks of RCL and RHCL volatile oils. Among them, peak 36 (zingiberenol), peak 37 (zingiberenol isomer), peak 40 (turmerone), and peak 42 (*β*-turmerone) showed similar peak ratios among all batches. The relative contents of peak 18 (caryophyllene) in RCL volatile oils were higher than that in RHCL, while peak 28 (*β*-sesquiphellandrene) and peak 27 (*β*-bisabolene) were on the contrary.

The similarity is an important parameter for the fingerprint analysis. It has been demonstrated that samples with similar fingerprints may have similar properties. In this case, the reference chromatogram (R) was generated based on the chromatograms of 12 samples, and the similarities of 12 different samples were evaluated by comparing each sample’s chromatogram with the reference chromatogram. Additionally, their values expressed 0.943–0.998, which indicated that these samples had high similarities (Table 3).

### 2.3. Profiling the Differences between RCL and RHCL Volatile Oil Using PCA and PLS-DA

Although there was a high degree of the similarities, there were some minor peaks that were different between RCL and RHCL. To evaluate the quality variation and difference between RCL and RHCL, PCA was performed based on the peak area ratio of 45 components. As shown in Figure 4A, the RCL and RHCL samples were separated from each other with the R^2^X at 0.658. These results demonstrated that volatile components of RCL and RHCL had some minor differences, which were hard to indicate using chromatographic fingerprints.

The PLS-DA technique was introduced to obtain better group clustering and discover the compounds’ differences. As shown in Figure 4B, the RCL and RHCL samples were clearly classified into two groups according to their original parts. The values of R^2^Y and Q^2^Y were 0.982 and 0.895, respectively, which indicated that the PLS-DA model was stable and had a better prediction. Based on the PLS-DA, the variable importance for the projection (VIP) plot was established to show the contribution of each variable to the discrimination of the RCL and RHCL samples. As shown in Figure 4C, 20 compounds with VIP values of more than 1 were screened as the potentially differential components among the RCL and RHCL samples. Among them, peak 38 (2-Hepten-1-ol,2-methyl-6-(4-methyl-1,4-cyclohexadien-1-yl)-,(2*Z*,6*R*)-(9CI,ACI)), peak 22 (humulene), peak 18 (caryophyllene), peak 14 ((−)-terpinen-4-ol), and peak 8 (eucalyptol) possessed the top five VIP values.

### 2.4. Antioxidant Activity of Volatile Oils by FRAP Assay

The sesquiterpenoids of *Curcuma* plants are one of the major groups of antioxidants besides curcuminoids. Therefore, the antioxidant activities of the volatile oils extracted from RCL and RHCL were evaluated based on their ferric-reducing antioxidant power. The results are listed in Table 4. The FRAP values of the volatile oils varied from 33.4 ± 21.4 to 438.4 ± 52.3 mM FeSO_4_/mL for RCL, while 30.8 ± 15.5 to 335.9 ± 16.7 mM/mL FeSO_4_ for RHCLs. The mean FRAP value of the six batches of RCL and RHCL samples was 265.5 and 149.8, respectively. Although the volatile oils yield of RCL was lower than RHCL, the antioxidant activity of the RCL volatile oils showed slightly better than RHCL’s. As previously reported [43], the essential oils of *C. longa* rhizomes gave the highest antioxidant activity than other medicinal rhizomes from *Curcuma*, and *α*-turmerone, *β*-turmerone, and *β*-sesquiphellandrene were determined as major contributing sesquiterpenoids. According to our GC–MS results, turmerone, *β*-turmerone, and *β*-sesquiphellandrene covered 59.64% of the RCL volatile components, which were slightly higher than those in RHCL (54.72%), calculated as the sum of their peak area ratio on average. However, the antioxidants from *C. longa* should be verified by reference standards in the future.

## 3. Materials and Methods

### 3.1. Materials, Reagents, and Reference Standards

The herbal samples including 6 batches of RCL (S1–S6) and 6 batches of RHCL (S7–S12) were collected from Sichuan province, China (Table 5). The species were identified by Professor Ping Li (China Pharmaceutical University, Nanjing, China), and the voucher specimens were stored in the State Key Laboratory of Natural Medicines, China Pharmaceutical University, Nanjing, China.

Anhydrous ethanol (GC grade) was bought from Yonghua Chemical Reagent Co., Ltd. (Suzhou, China). Anhydrous sodium sulfate was obtained from Nanjing Chemical Reagent Co., Ltd. (Nanjing, China). Ultrapure water was prepared by Milli-Q water purification system (Millipore, Bedford, MA, USA). The commercial kits for ferric-reducing antioxidant power (FRAP) assay were purchased from Beyotime Institute of Biotechnology (Shanghai, China).

Reference standards of (−)-terpinen-4-ol, caryophyllene oxide, (*E*)-*β*-farnesene were bought from Shanghai Yuanye Bio-Technology Co., Ltd. (Shanghai, China). *N*-tridecane, *α*-terpineol, *β*-caryophyllene, *α*-terpinene, Linalool, and *m*-cymene were obtained from Push Bio-technology Co., Ltd. (Chengdu, China). Terpinolene was bought from Shanghai TCI Development Co., Ltd. (Shanghai, China). 3-Carene was obtained from Rhawn Reagent (Shanghai, China). *D*-Limonene was bought from Shanghai Sigma-Aldrich Co., Ltd. (Shanghai, China). Eucalyptol was obtained from National Institute for Food and Drug Control (Beijing, China). *ar*-Turmerone was bought from BioBioPha Co., Ltd. (Yunnan, China). *p*-Cymen-8-ol was obtained from Shanghai Zzbio Co., Ltd. (Shanghai, China). All the purities of the reference standards mentioned above were not less than 80% (GC).

### 3.2. Volatile Oil Extraction and Sample Preparation

The dried RCL and RHCL samples were pulverized and griddled through 24 mesh. Accurately weighed 40 g powder of each sample was extracted with 320 mL water in a Clevenger-type apparatus for 5 h. The extracted volatile oil was dried with anhydrous sodium sulfate and stored at −20 °C. The extraction yield was calculated in a milliliter of oil per 40 g of dried RCL and RHCL.

A solution of *N*-tridecane, an internal standard (IS), was prepared in anhydrous ethanol at a concentration of 1 mg/mL. Then, this solution was stored at −80 °C. For GC–MS analysis, the extracted 1 μL of volatile oil was transferred to a centrifuge tube and diluted 1000-fold by anhydrous ethanol. A certain amount of *N*-tridecane was also added into the sample with the final concentration at 10 μg/mL. Then, the sample solution was filtered through a 0.22 μm syringe filter before injection.

### 3.3. Chemical Profiling of the RCL and RHCL Volatile Oils

#### 3.3.1. GC–MS Condition

GC–MS was performed on an Agilent 7890B GC system equipped with Agilent 5977A Mass Selective Detector (Agilent Technologies, Santa Clara, CA, USA).

Chromatographic separation was performed on an Agilent DB-5 column (60 m × 0.25 mm, 0.25 μm). Helium was used as carrier gas with a constant flow rate of 1.0 mL/min. The oven temperature program was started at 60 °C, increased to 100 °C at a rate of 5 °C/min, increased to 140 °C at a rate of 10 °C/min, held for 2 min, and increased to 155 °C at a rate of 1 °C/min, held for 2 min, increased to 160 °C at a rate of 0.5 °C/min, increased to 180 °C at a rate of 3 °C/min, and finally reached 260 °C at a rate of 15 °C/min. Each 1 μL aliquot of volatile oil sample solution was injected into the GC–MS system at a split ratio of 10:1.

The electron impact ionization mode at 70 eV was used. The temperature of the ion source and MS quadrupole were set to 230 °C and 150 °C. The filament on delay was set for 9 min, and the runtime was from 9 to 53 min. All data were acquired in a full-scan mode within a mass range from m/z 50–600.

Data were acquired by Agilent MassHunter Acquisition Software version B.06.01 (Agilent Technologies, Santa Clara, CA, USA). Data analysis was performed by Agilent MassHunter Qualitative Analysis Software version B.10.00 (Agilent Technologies, Santa Clara, CA, USA), and all components were identified by comparing their mass fragments with the standard mass spectra from NIST Mass Spectral Search Program version 14.0 (National Institute of Standards and Technology, MD, USA).

#### 3.3.2. Methodology Validation of Fingerprint Analysis

The analytical method was validated for precision, repeatability, and stability. Based on the established GC–MS condition programs, we repeatedly injected the same sample solution six times to evaluate the instrumental run-to-run precision. The stability was tested with the same sample solution at room temperature and analyzed at 0, 2, 4, 8, 12, 20, and 24 h, injecting independent sample solutions from the same batch of volatile oil sample (S6) to assess method repeatability. The similarities among chromatograms were calculated to assess the precision, repeatability, and stability. Additionally, the relative peak area and the relative retention time of 11 characteristic chromatographic peaks were selected for the assisted methodological validation.

#### 3.3.3. Similarity Analysis of Fingerprints

The GC–MS data (including peak areas and retention times) were exported from MassHunter Qualitative Analysis Software version B.10.00 (Agilent Technologies, Santa Clara, CA, USA) as “.txt” document. In addition, peaks whose peak area accounted for no less than 0.1% of the total peak area participated in the matching. Then, these data were imported into the Similarity Evaluation System for Chromatographic Fingerprint of Traditional Chinese Medicine version 2004 A (Chinese Pharmacopoeia Commission, Beijing, China). Fingerprints were matched automatically and established. The reference fingerprint (R) was generated with the median method, and the similarity values between the entire chromatographic profiles of volatile oil samples and the reference fingerprint (R) were calculated.

### 3.4. Principle Component Analysis and Partial Least Squares Discriminant Analysis

The principal component analysis (PCA) was applied for analyzing the correlation of samples by reducing the dimensions of the original data. In this work, the peak area ratio of 45 components in volatile oils of 12 batches of samples were imported into the R software version 4.1.2 (R Core Team, Vienna, Austria) for conducting PCA.

Partial least squares discriminant analysis (PLS-DA), a supervised method, was introduced to group classification. It takes advantage of class information to attempt to maximize classification. In this work, we used the same data as input data to perform PLS-DA for RCL and RHCL volatile oil samples using R software version 4.1.2 (R Core Team, Vienna, Austria). The variable importance for the projection (VIP) values were employed to find chemical constituents that can help discriminate different samples.

### 3.5. Total Antioxidant Capacity Assay with FRAP Method

The FRAP assay was carried out by following the instruction manual. Working standards were obtained by diluting 10 mM FeSO_4_·7H_2_O with anhydrous ethanol. A series of solutions with known FeSO_4_·7H_2_O concentration (0.15, 0.3, 0.6, 0.9, 1.2, and 1.5 mM) were used for calibration. Volatile oil was diluted with anhydrous ethanol. Then, 5 μL of volatile oil sample solution (containing 3 μL of volatile oil) was mixed with 180 μL of newly prepared FRAP reagent. After incubation at 37 °C for 5 min, the absorbance of each mixture was monitored at 593 nm. Results were expressed as reduced mM FeSO_4_ per milliliter volatile oil. All determinations were conducted in triplicates, and the mean ± standard deviation values were finally expressed.

## 4. Conclusions

In this study, the volatile oils in RCL and RHCL samples were extracted by hydrodistillation and analyzed by GC–MS. The chemical constituents in the volatile oils were characterized based on a mass spectra database, previous literature, and reference standards. In addition, these components mainly included monoterpenes and sesquiterpenes. Among them, turmerone, (−)-zingiberene, and *β*-turmerone were dominant compounds detected in both RCL and RHCL. The GC–MS fingerprints showed the holistically chromatographic similarities among sample batches, whereas chemometrics revealed the minor chemical differences for discrimination of the RCL and RHCL samples. Moreover, the FRAP results indicated that the RCL and RHCL had slightly different antioxidant levels, which deserved attention for screening the related antioxidative volatile compounds in future studies. To some extent, this study complemented the material basis of RCL and RHCL, which could facilitate further pharmacological research and quality control.

## Figures and Tables

**Figure 1 molecules-27-03196-f001:**
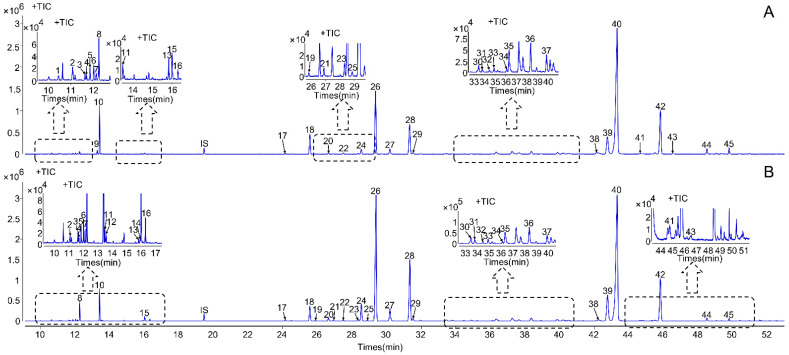
The GC–MS total ion chromatograms of the RCL (**A**) and RHCL (**B**) volatile oils (peak IS represented the internal standard).

**Figure 2 molecules-27-03196-f002:**
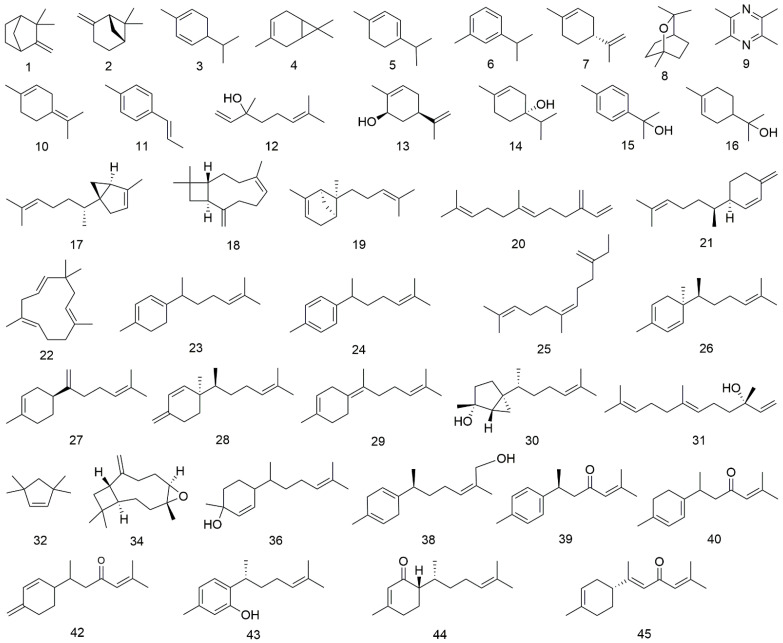
Chemical structures of the compounds identified from RCL and RHCL.

**Figure 3 molecules-27-03196-f003:**
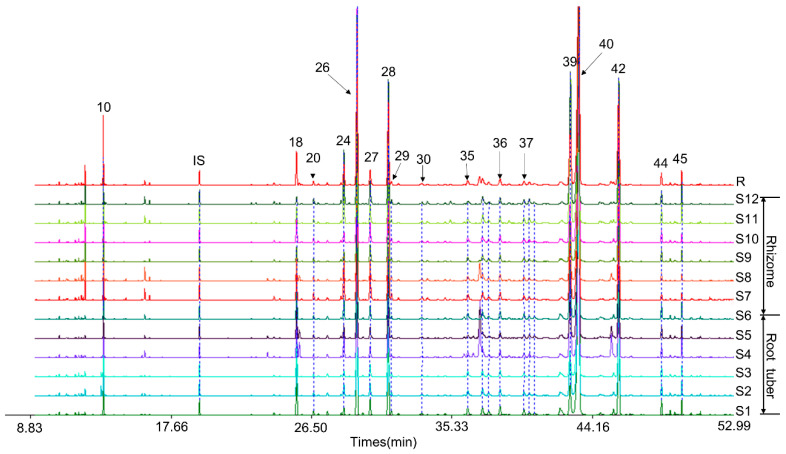
The GC–MS fingerprints of RCL and RHCL volatile oils. Chromatograms S1–S6 represented RCL volatile oils, whereas S7–S12 represented RHCL volatile oils. R was the reference chromatogram. Peak IS was the internal standard (*N*-tridecane). Peaks 10, 18, 20, 24, 26, 27, 28, 29, 30, 35, 36, 37, 39, 40, 42, 44, and 45 were seventeen common peaks.

**Figure 4 molecules-27-03196-f004:**
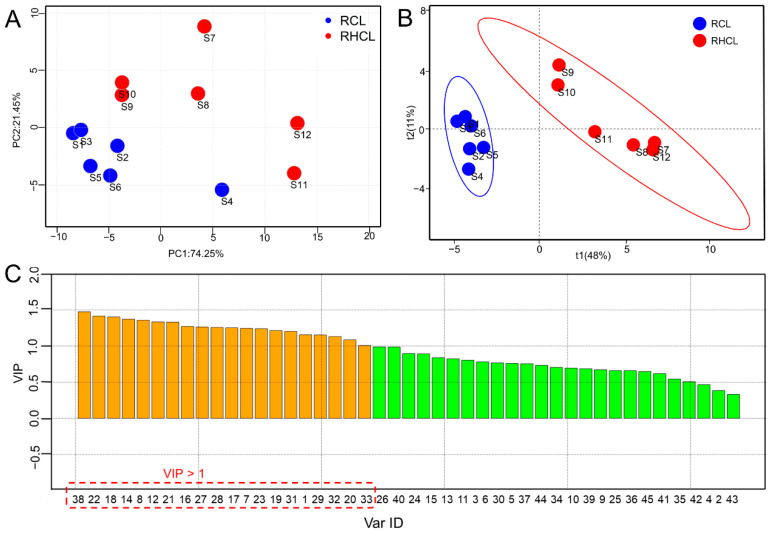
Two-dimensional PCA (**A**) and PLS-DA (**B**) score plot of RCL and RHCL volatile oil samples. The VIP plot (**C**) obtained from PLS-DA.

**Table 1 molecules-27-03196-t001:** Chemical constituents in RCL and RHCL volatile oils were identified by GC–MS.

Peak No.	RT (min)	Identification	Formula	RCL		RHCL	
				^#^ Area Ratio (%)	Average (%)	^#^ Area Ratio (%)	Average (%)
1	10.44	Camphene	C_10_H_16_	0.01–0.05	0.03	N.D.	/
2	11.09	(−)-*β*-Pinene	C_10_H_16_	0.02–0.12	0.07	0.02–0.11	0.05
3	11.63	*α*-Phellandrene	C_10_H_16_	0.03–0.08	0.06	0.02–0.08	0.06
*4	11.70	3-Carene	C_10_H_16_	0.01–0.04	0.02	0.02–0.04	0.03
*5	11.85	α-Terpinene	C_10_H_16_	0.03–0.12	0.08	0.04–0.11	0.09
*6	12.02	*m*-Cymene	C_10_H_14_	0.02–0.08	0.05	0.02–0.18	0.09
*7	12.14	*D*-Limonene	C_10_H_16_	0.01–0.04	0.03	0.05–0.10	0.07
*8	12.26	Eucalyptol	C_10_H_18_O	0.07–0.25	0.13	0.53–1.39	1.02
9	13.27	Pyrazine, tetramethyl-	C_8_H_12_N_2_	0–0.35	0.09	N.D.	/
*10	13.40	Terpinolene	C_10_H_16_	1.31–4.07	2.67	0.88–2.57	1.93
11	13.50	*p*-(1-Propenyl)-toluene	C_10_H_12_	0.03–0.06	0.04	0.02–0.08	0.06
*12	13.58	Linalool	C_10_H_18_O	N.D.	/	0.03–0.07	0.04
13	15.84	(−)-*cis*-Carveol	C_10_H_16_O	0–0.09	0.04	0–0.02	0.01
*14	15.94	(−)-Terpinen-4-ol	C_10_H_18_O	N.D.	/	0.01–0.03	0.02
*15	16.02	*p*-Cymen-8-ol	C_10_H_14_O	0.03–0.31	0.14	0.02–0.58	0.27
*16	16.31	*α*-Terpineol	C_10_H_18_O	0.01–0.04	0.02	0.06–0.16	0.13
17	24.14	7-*epi*-Sesquithujene	C_15_H_24_	0.08–0.10	0.09	0.12–0.19	0.15
*18	25.58	Caryophyllene	C_15_H_24_	3.14–3.75	3.42	0.49–2.12	1.43
19	25.88	*cis*-*α*-Bergamotene	C_15_H_24_	0.02–0.03	0.03	0.04–0.06	0.05
*20	26.63	(*E*)-*β*-Farnesene	C_15_H_24_	0.19–0.27	0.24	0.25–0.42	0.34
21	26.92	(−)-*β*-Sesquiphellandrene	C_15_H_24_	0.05–0.07	0.06	0.08–0.11	0.10
22	27.51	Humulene	C_15_H_24_	0.22–0.24	0.23	0.03–0.14	0.10
23	28.36	*γ*-Curcumene	C_15_H_24_	0.08–0.15	0.11	0.16–0.26	0.20
24	28.52	*α*-Curcumene	C_15_H_22_	0.48–2.22	1.11	0.81–3.85	2.28
25	28.88	(*E*)-*β*-Farnesene isomer	C_15_H_24_	0.04–0.11	0.06	0.05–0.11	0.08
26	29.38	(−)-Zingiberene	C_15_H_24_	8.74–14.26	11.99	11.22–21.15	16.20
27	30.22	*β*-Bisabolene	C_15_H_24_	0.96–1.15	1.06	1.44–2.13	1.69
28	31.33	*β*-Sesquiphellandrene	C_15_H_24_	5.32–6.77	6.17	7.92–10.88	9.06
29	31.51	*trans*-*γ*-Bisabolene	C_15_H_24_	0.22–0.31	0.28	0.31–0.46	0.37
30	33.46	*cis*-Sesquisabinene hydrate	C_15_H_26_O	0.13–0.22	0.17	0.15–0.25	0.21
31	33.78	Nerolidol	C_15_H_26_O	0.03–0.08	0.05	0.10–0.26	0.17
32	34.47	3,3,5,5-Tetramethylcyclopentene	C_9_H_16_	0.03–0.04	0.04	0.04–0.08	0.06
33	34.90	Tumerone isomer	C_15_H_22_O	0.08–0.10	0.09	0.09–0.16	0.13
*34	36.11	Caryophyllene oxide	C_15_H_24_O	0.03–0.34	0.13	0.01–0.13	0.06
35	36.35	*cis*-Sesquisabinene hydrate or isomer	C_15_H_26_O	0.38–0.78	0.64	0.46–0.61	0.56
36	38.37	Zingiberenol	C_15_H_26_O	0.60–0.85	0.76	0.63–0.77	0.70
37	39.88	Zingiberenol isomer	C_15_H_26_O	0.41–0.55	0.48	0.34–0.51	0.43
38	42.13	2-Hepten-1-ol,2-methyl-6-(4-methyl-1,4-cyclohexadien-1-yl)-,(2*Z*,6*R*)-(9CI,ACI)	C_15_H_24_O	0.72–1.03	0.89	0.33–0.57	0.44
*39	42.8	*ar*-Turmerone	C_15_H_20_O	3.24–10.28	5.58	4.23–16.54	8.93
40	43.25	Turmerone	C_15_H_22_O	33.48–46.19	42.78	28.94–41.73	35.18
41	44.81	*cis*-Sesquisabinene hydrate or isomer	C_15_H_26_O	0.08–0.25	0.15	0.08–0.16	0.11
42	45.84	*β*-Turmerone	C_15_H_22_O	9.92–11.32	10.69	8.02–12.57	10.48
43	46.54	Curcuphenol	C_15_H_22_O	0.07–0.17	0.12	0.04–0.21	0.12
44	48.52	(6*R*, 7*R*)-Bisabolone	C_15_H_24_O	0.83–1.03	0.90	0.40–1.00	0.76
45	49.80	(*E*)-Atlantone	C_15_H_22_O	0.63–1.33	0.81	0.26–1.01	0.60
Total			/	/	92.60	/	94.86

*: Identified with the reference standards; **^#^**: The ranges of six investigated batches; “N.D.”: not detected.

**Table 2 molecules-27-03196-t002:** Precisions, stabilities, and respectabilities of the GC–MS method.

Peak No.	Precision (*n* = 6, RSD, %)		Stability (*n* = 6, RSD, %)		Repeatability (*n* = 6, RSD, %)	
	RPA	RRT	RPA	RRT	RPA	RRT
10	1.76	<1.00 × 10^−3^	3.46	<1.00 × 10^−3^	4.59	2.66 × 10^−2^
18	0.47	2.02 × 10^−2^	2.28	1.48 × 10^−2^	4.80	3.42 × 10^−2^
24	0.63	1.43 × 10^−2^	2.61	<1.00 × 10^−3^	5.06	3.40 × 10^−2^
26	0.85	1.76 × 10^−2^	1.38	1.29 × 10^−2^	4.20	3.54 × 10^−2^
27	1.44	2.49 × 10^−2^	1.00	3.15 × 10^−2^	4.96	2.78 × 10^−2^
28	0.95	<1.00 × 10^−3^	1.30	<1.00 × 10^−3^	4.04	2.09 × 10^−2^
39	1.03	9.54 × 10^−3^	1.70	1.84 × 10^−2^	4.88	3.10 × 10^−2^
40	0.78	9.42 × 10^−3^	1.77	2.47 × 10^−2^	4.26	3.18 × 10^−2^
42	0.81	1.13 × 10^−2^	1.74	1.17 × 10^−2^	4.23	3.06 × 10^−2^
44	1.75	1.84 × 10^−2^	2.86	1.10 × 10^−2^	5.07	2.96 × 10^−2^
45	4.68	1.04 × 10^−2^	3.65	1.07 × 10^−2^	3.90	2.66 × 10^−2^

**Table 3 molecules-27-03196-t003:** The chromatographic similarities with GC–MS fingerprints for RCL and RHCL volatile oils.

Samples ID	Similarity	Samples ID	Similarity
S1	0.996	S7	0.965
S2	0.998	S8	0.989
S3	0.997	S9	0.996
S4	0.971	S10	0.994
S5	0.993	S11	0.943
S6	0.997	S12	0.943

**Table 4 molecules-27-03196-t004:** The results of FRAP assay.

Sample ID	FRAP Values (Mean ± SD, *n* = 3, mM FeSO_4_/mL)	Sample ID	FRAP Values (Mean ± SD, *n* = 3, mM FeSO_4_/mL)
S1	438.4 ± 52.3	S7	30.8 ± 15.5
S2	207.2 ± 30.0	S8	51.1 ± 18.1
S3	397.3 ± 21.8	S9	210.7 ± 21.3
S4	224.4 ± 17.3	S10	158.6 ± 10.7
S5	33.4 ± 21.4	S11	335.9 ± 16.7
S6	292.1 ± 4.1	S12	111.7 ± 12.7

**Table 5 molecules-27-03196-t005:** Basic information of RCL and RHCL samples.

Sample ID	Medicinal Parts	Source
S1	Root tuber	Shuangliu, Chengdu, Sichuan
S2	Root tuber	Shuangliu, Chengdu, Sichuan
S3	Root tuber	Shuangliu, Chengdu, Sichuan
S4	Root tuber	Qianwei, Leshan, Sichuan
S5	Root tuber	Qianwei, Leshan, Sichuan
S6	Root tuber	Qianwei, Leshan, Sichuan
S7	Rhizome	Shuangliu, Chengdu, Sichuan
S8	Rhizome	Shuangliu, Chengdu, Sichuan
S9	Rhizome	Shuangliu, Chengdu, Sichuan
S10	Rhizome	Shuangliu, Chengdu, Sichuan
S11	Rhizome	Qianwei, Leshan, Sichuan
S12	Rhizome	Qianwei, Leshan, Sichuan

## Data Availability

Not applicable.

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
