# Peer review of "Comparative Analysis of Volatile Constituents in Root Tuber and Rhizome of Curcuma longa L. Using Fingerprints and Chemometrics Approaches on Gas Chromatography–Mass Spectrometry"

_molecules, 2022, doi:10.3390/molecules27103196_

Round 1

Reviewer 1 Report

In general, the design of the assays are correct and the analysis and identification of compounds appropriate. The only comment regarding the analysis of results is that, although the authors applied the type of statistical methodologies usually used in type of data, the values due to their generation with respect to the standard (or total area of peaks) do not belong to the Euclidean space. The multivariate methods used were developed for this sampling space and then, at least formally, are not appropriate. Authors should investigate methods available for the simplex, the sampling space of their data.

However, this problem does not mean any negative consideration with respect to the publication of the manuscript since the tools used in it are those applied by most authors.

Another question is the average of %. How were they estimated? Standard statistics are not appropriate for working with proportions.

Author Response

Summarized Comment: In general, the design of the assays are correct and the analysis and identification of compounds appropriate. The only comment regarding the analysis of results is that, although the authors applied the type of statistical methodologies usually used in type of data, the values due to their generation with respect to the standard (or total area of peaks) do not belong to the Euclidean space. The multivariate methods used were developed for this sampling space and then, at least formally, are not appropriate. Authors should investigate methods available for the simplex, the sampling space of their data.

            However, this problem does not mean any negative consideration with respect to the publication of the manuscript since the tools used in it are those applied by most authors.

Response: We feel great thanks for your professional review work on our manuscript. In this study, a 45-dimensional vector composed of 45 peak areas is used to represent a sample, and the Euclidean distance between two samples can be calculated. PLS-DA can reflect the importance of each variable by calculating the VIP value. It is used to describe the contribution of each component to the distinction between RCL and RHCL. Thank you again.

Detailed Comment 1: Another question is the average of %. How were they estimated? Standard statistics are not appropriate for working with proportions.

Response: Thank you for your professional suggestion. In this work, we collected and analyzed 6 batches of RCL (S1-S6) and 6 batches of RHCL (S7-S12), respectively. The peak ratio of the analyte in each sample was calculated based on the area normalization method. The average of % means the average of analyte’s area ratio in 6 batches of RCL volatile oils or 6 batches of RHCL volatile oils, respectively. 

Reviewer 2 Report

This work reveals the volatile constituents in root tuber and rhizome of Curcuma longa L, which provides new knowledge about Curcuma longa L and has certain contribution for further development. Reliable methods of instrumental analysis and statistical analysis are used. The conclusions are supported by the data. However, the authors should take care of the following aspects to improve their manuscript.

  • The abstract can be more concise. Some processing description is not required.
  • Grammatical problems should be carefully corrected, e.g., “There were also some compounds were varied in RCL and RHCL”, “to analysis”....
  • For VIP, full name should be given once the abbreviation first appears.
  • The results are lack of discussion, especially with other reports.

Author Response

We deeply appreciate the consideration and comments from Reviewer 2 on our manuscript “Molecules-1684283”.

Summarized Comment: This work reveals the volatile constituents in root tuber and rhizome of Curcuma longa L, which provides new knowledge about Curcuma longa L and has certain contribution for further development. Reliable methods of instrumental analysis and statistical analysis are used. The conclusions are supported by the data. However, the authors should take care of the following aspects to improve their manuscript.

Response: Thank you for your review and valuable comments on our manuscript. We have revised our manuscript by fully considering the suggestions and adding the necessary content.

Detailed Comment 1: The abstract can be more concise. Some processing description is not required.

Response: Thank you for your suggestion. We have revised the abstract to be concise. All revisions were marked in the revised paper.

Detailed Comment 2: Grammatical problems should be carefully corrected, e.g., “There were also some compounds were varied in RCL and RHCL”, “to analysis”....

Response: Thank you for your suggestion. We have corrected accordingly. The listed sentences were rewritten, which were revised as “There were also some compounds varied in RCL and RHCL”, and “Therefore, the volatile oils of RCL and RHCL were extracted by hydrodistillation and then analyzed by GC-MS”. Moreover, we have carefully polished the language of the full paper.

Detailed Comment 3: For VIP, full name should be given once the abbreviation first appears.

Response: Thank you for your kind comments. The full name of VIP, Variable Importance for the Projection, was added where it first appears (in Abstract section).

Detailed Comment 4: The results are lack of discussion, especially with other reports.

Response: Thanks for your comments. We have added some discussions in our manuscript such as “We found that all of the samples in our research belonged to the reported major chem-ical type of turmeric in Xu's article [41]”, “As the previous report [42], the essential oils of C. longa rhizomes gave the highest antioxidant activity than other medicinal species from Curcuma. Furthermore, α-turmerone, β-turmerone and β-sesquiphellandrene were determined as major contributed sesquiterpenoids…” and so on. All revisions were marked in red.

Related references:

[41]     Xu, L.L.; Shang, Z.P.; Lu, Y.Y.; Li, P.; Sun, L.; Guo, Q.L.; Bo, T.; Le, Z.Y.; Bai, Z.L.; Zhang, X.L.; Qiao, X.; Ye, M. Analysis of curcuminoids and volatile components in 160 batches of turmeric samples in China by high-performance liquid chromatography and gas chromatography mass spectrometry. J Pharm Biomed Anal 2020, 188, 1-7.

[42] Zhao, J.; Zhang, J.S.; Yang, B.; Lv, G.P.; Li, S.P. Free radical scavenging activity and characterization of sesquiterpenoids in four species of Curcuma using a TLC bioautography assay and GC-MS analysis. Molecules 2010, 15, 7547-7557.

Reviewer 3 Report

In the manuscript molecules-1684283 the oils extracted from Curcuma longa L. root tuber (RCL) and rhizome (RHCL) were characterised by gas chromatography-mass spectrometry (GC-MS). The obtained GC-MS profiles were then subjected to principal component analysis (PCA) and partial least squares regression-discriminant analysis (PLS-DA). Therefore, this study is valuable to address Curcuma longa L. volatile oils characterisation, although major modifications must be made.  

This is an interesting study dealing with Curcuma longa L. volatile compounds. However, the manuscript is not adequate for publication in its current form since many sentences need to be grammatically corrected. Therefore, I encourage the authors to correct it.  

Line 54. Change “Gas chromatography-mass spectrometer” to “Gas chromatography-mass spectrometry”.

In Table 2, “N.A.” is defined, but it is not used in the Table.  

Regarding method validation, the authors claimed to validate the precision, repeatability, and stability of the analytical method. In analytical chemistry, repeatedly injecting the same sample solution correspond to the “instrumental run-to-run precision” or “instrumental repeatability”, but not just “precision”. Instead, injecting independent sample solutions corresponds to “method repeatability” not just “repeatability”.

Moreover, in my opinion, the method validation lacks relevant parameters such as “day-to-day precision” to be completed. Therefore, I suggest calling it “Quality parameters evaluation” and not “Method validation”.    

Please, provide more details about the similarity analysis of GC-MS fingerprints. For instance, how does the median method work, and how are the similarity values calculated?

Have the compounds identified in this study been found in previous works? Please, provide this information and discuss it in Section 2.2.  

Do the 17 compounds identified after the similarity analysis present a similar content in RCL and RHCL volatile oils?

Line 155 to 157. I would not conclude that the better in-group clustering of RCL samples is due to their quality since many external conditions can influence this fact.  

Line 262. Remove “is a regression extension of PCA that”

Author Response

We deeply appreciate the consideration and comments from Reviewer 3 on our manuscript “Molecules-1684283”.

Summarized Comment: In the manuscript molecules-1684283 the oils extracted from Curcuma longa L. root tuber (RCL) and rhizome (RHCL) were characterised by gas chromatography-mass spectrometry (GC-MS). The obtained GC-MS profiles were then subjected to principal component analysis (PCA) and partial least squares regression-discriminant analysis (PLS-DA). Therefore, this study is valuable to address Curcuma longa L. volatile oils characterisation, although major modifications must be made.

            This is an interesting study dealing with Curcuma longa L. volatile compounds. However, the manuscript is not adequate for publication in its current form since many sentences need to be grammatically corrected. Therefore, I encourage the authors to correct it.

Response: Thank you for your review and valuable comments on our manuscript. We have revised our manuscript by fully considering all of reviewer’s suggestions in detail.

Detailed Comment 1: Line 54. Change “Gas chromatography-mass spectrometer” to “Gas chromatography-mass spectrometry”.

Response: Thank you for your suggestion. We have revised “Gas chromatography-mass spectrometer” asGas chromatography-mass spectrometry.

Detailed Comment 2: In Table 2, “N.A.” is defined, but it is not used in the Table.

Response: Thanks for your comments. We have corrected it accordingly.

Detailed Comment 3: Regarding method validation, the authors claimed to validate the precision, repeatability, and stability of the analytical method. In analytical chemistry, repeatedly injecting the same sample solution correspond to the “instrumental run-to-run precision” or “instrumental repeatability”, but not just “precision”. Instead, injecting independent sample solutions corresponds to “method repeatability” not just “repeatability”.

Response: Thank you for your suggestion. The method validation described in our manuscript refers to the fingerprint method validation reported in most articles [1-4]. Then according to your nice comments, we have made changes in Section 3.3.2, as follows: “we repeatedly injected the same sample solution six times to evaluate the instrumental run-to-run precision”, “Injecting six independent sample solutions from the same batch of volatile oil sample (S6) to assess method repeatability”.

Related references:

[1] Zhang D, Fan L, Yang N, et al. Discovering the main "reinforce kidney to strengthening Yang" active components of salt Morinda officinalis based on the spectrum-effect relationship combined with chemometric methods [J]. J Pharm Biomed Anal. 2022, 207: 114422.

[2] Zhang J, Gong D, Lan L, Zheng Z, Pang X, Guo P, Sun G. Comprehensive evaluation of Loblolly fruit by high performance liquid chromatography four wavelength fusion fingerprint combined with gas chromatography fingerprinting and antioxidant activity analysis [J]. J Chromatogr A. 2022, 1665: 462819.

[3] Wang LJ, Jiang ZM, Xiao PT, Sun JB, Bi ZM, Liu EH. Identification of anti-inflammatory components in Sinomenii Caulis based on spectrum-effect relationship and chemometric methods [J]. J Pharm Biomed Anal. 2019, 167: 38-48.

[4] Wang Y, He T, Wang J, Wang L, Ren X, He S, Liu X, Dong Y, Ma J, Song R, Wei J, Yu A, Fan Q, Wang X, She G. High performance liquid chromatography fingerprint and headspace gas chromatography-mass spectrometry combined with chemometrics for the species authentication of Curcumae Rhizoma [J]. J Pharm Biomed Anal. 2021, 202: 114144.

Detailed Comment 4: Moreover, in my opinion, the method validation lacks relevant parameters such as “day-to-day precision” to be completed. Therefore, I suggest calling it “Quality parameters evaluation” and not “Method validation”.

Response: Thanks for your comment. We have changed “Method validation” to “Methodology validation of fingerprint analysis”.

Detailed Comment 5: Please, provide more details about the similarity analysis of GC-MS fingerprints. For instance, how does the median method work, and how are the similarity values calculated?

Response: Thanks for your comment. The similarity analysis of GC-MS fingerprints was carried out by the software approved by the Chinese Pharmacopoeia Commission, named Similarity Evaluation System for Chromatographic Fingerprint of Traditional Chinese Medicine (version 2004 A; Beijing, China). In this software, the similarity values could be automatically calculated. Meanwhile, the reference fingerprint (R) could be generated with the median method or mean value method. The median method arranges the corresponding vectors according to their size and takes the vector in the middle as the reference model. If the number of samples is odd, there is only one median; When the number of samples is even, the median is the average of the two numbers.

Detailed Comment 6: Have the compounds identified in this study been found in previous works? Please, provide this information and discuss it in Section 2.2.

Response: Thank you for your comments. We identified the compounds by using the reference standards and also based on the previous reports [6,34,38-40]. For example, α-Phellandrene, α-Terpinene, Eucalyptol, p-Cymen-8-ol, 7-epi-Sesquithujene, Caryophyllene, cis-α-Bergamotene, (E)-β-Farnesene, α-Curcumene, (-)-Zingiberene, β-Bisabolene, β-Sesquiphellandrene, Caryophyllene oxide, Zingiberenol, ar-Turmerone, Turmerone, β-Turmerone, (6R, 7R)-Bisabolone and (E)-Atlantone were also identified by Wang et al [38]. We have supplemented the reference and added some discussions in our revised manuscript.

Related references:

  1. Chen, Z.M.; Quan, L.; Zhou, H.T.; Zhao, Y.F.; Chen, P.; Hu, L.; Yang, Z.; Hu, C.J.; Cao, D. Screening of active fractions from Curcuma Longa Radix isolated by HPLC and GC-MS for promotion of blood circulation and relief of pain. J Ethnopharmacol 2019, 234, 68-75.
  2. Hu, Y.C.; Kong, W.J.; Yang, X.H.; Xie, L.W.; Wen, J.; Yang, M.H. GC-MS combined with chemometric techniques for the quality control and original discrimination of Curcumae longae rhizome: analysis of essential oils. J Sep Sci 2014, 37, 404-411.

38        Wang, L.; Li. X.; Wang, Y.; Ren, X.; Liu, X.; Dong, Y.; Ma, J.; Song, R.; Wei, J.; Yu, A.; Fan, Q.; Shan, D.; Yao, J.; She, G. Rapid discrimination and screening of volatile markers for varietal recognition of Curcumae Radix using ATR-FTIR and HS-GC-MS combined with chemometrics. J Ethnopharmacol 2021, 280, 1-10.

  1. Xu, F.X.; Zhang, J.Y.; Jin, J; Li, Z.G.; She, Y.B.; Lee, M.R. Microwave-assisted natural deep eutectic solvents pretreatment followed by hydrodistillation coupled with GC-MS for analysis of essential oil from turmeric (Curcuma longa L.). J Oleo Sci 2021, 70, 1481-1494.
  2. Zhang, L.Y.; Yang, Z.W.; Chen, F.; Su, P.; Chen, D.K.; Pan, W.Y.; Fang, Y.X.; Dong, C.Z.; Zheng, X.; Du, Z.Y. Composition and bioactivity assessment of essential oils of Curcuma longa L. collected in China. Ind Crops Prod 2017, 109, 60-73.

Detailed Comment 7: Do the 17 compounds identified after the similarity analysis present a similar content in RCL and RHCL volatile oils?

Response: Thank you for your comments. The common peak refers to the peak that appears in all chromatograms, as defined by the software of Similarity Evaluation System. Then, 17 common peaks were found from the 12 analyzed samples. The relative contents of the compounds of Peak 36, Peak 37, Peak 40 and Peak 42 were similar among all batches. The relative contents of the compound of Peak 18 in RHCL volatile oils were higher than that in RCL volatile oils. The relative contents of the compounds of Peak 27 and Peak 28 were higher in RHCL than that in RCL volatile oils. The peak area ratio was listed in Table 1. We have supplemented the results and discussion in the revised manuscript.

Detailed Comment 8: Line 155 to 157. I would not conclude that the better in-group clustering of RCL samples is due to their quality since many external conditions can influence this fact.

Response: Thanks for your comments. We have carefully revised our manuscript and re-written this part. Thank you again.

Detailed Comment 9: Line 262. Remove “is a regression extension of PCA that”

Response: Thank you for your comments. We have removed it accordingly.

Round 2

Reviewer 3 Report

The authors improved their manuscript and answered all my questions and comments properly. Therefore, I suggest the acceptance of this manuscript for publication in molecules.